# Infrared spectroscopic characterization of sesamin, a dietary lignan natural product

**Sara W. Jackson**[1]◉, **Moon-Hyung Jang**◉[2]◉*, **Eliza Asani**[1], **C. Ryan Yates**[3], **Joseph Ng**[1], **Jerome Baudry**◉[1]*

1 Department of Biological Sciences, The University of Alabama in Huntsville, Huntsville, Alabama, United States of America, 2 Department of Chemical and Materials Engineering, The University of Alabama in Huntsville, Huntsville, Alabama, United States of America, 3 National Center for Natural Products Research, The University of Mississippi, Oxford, Mississippi, United States of America

◉ These authors contributed equally to this work.
* jerome.baudry@uah.edu (JB); mj0090@uah.edu (MHJ)

**Data Availability Statement:** The data can be accessed from the BioStudies repository, information is as follows: Accession #: S-BSST1518 DOI: 10.6019/S-BSST1518.

## Abstract

Sesamin, a lignan component of sesame seed oil, has shown pharmacologic benefits, such as anti-oxidative and anti-inflammatory qualities. However, the amount of data available to the field is surprisingly sparse, as for instance there is no known spectroscopic characterization of sesamin. This work provides the first experimental infrared spectrum of sesamin. Sesamin powder was subjected to experimental Fourier-transform infrared spectroscopy, and the resulting spectrum was compared to quantum chemical calculations of sesamin's stereoisomers in various hydration states. Major peaks of sesamin were assigned vibrational modes through comparison of computed and observed spectra. Multiple sesamin species may be present in a typical powder sample, coexisting with potential trace hydration.

## Introduction

Sesamin is one of the lignan components of sesame seed oil. It has been shown in both basic and clinical research studies to have various beneficial pharmacologic effects, including anti-inflammatory and anti-oxidative properties [1–4]. Sesamin's role in various cellular signaling pathways has been proposed, although specific mechanisms of action have yet to be described [5, 6]. Proposals have been made in various reports to clinically utilize this fat-soluble compound to supplement or replace currently prescribed anti-inflammatory drugs to mitigate adverse side effects [7]. Despite its biologic and potential commercial importance, a physical description of sesamin outside of basic chemical characterizations has not yet been reported with the exception of structural crystallographic data with very broad and limited vibrational considerations [8, 9]. In general, as natural products' use in clinical applications is increasingly considered and regulated, methods of their characterization are ever important [10]. Recent spectroscopic characterization of sesamin content in sesame seeds from different locations has been described [11]. This work indicated that the sesamin content of the seeds varies with the geographical origin of the seeds. This illustrates the importance of vibrational spectroscopy in sesamin work, and demonstrates the need for detailed characterization of the peaks in IR fingerprint of sesamin. Hence, a complete analytical and structural description of sesamin is

**Funding:** The author(s) received no specific funding for this work.

**Competing interests:** The authors have declared that no competing interests exist.

needed to assist the continuing biological and clinical investigations into its mechanisms and efficacy as a modulator of inflammatory processes. One of the goals of this work is to take the first step toward an exhaustive spectral characterization of sesamin.

This article reports on the first sesamin peak assignment from Fourier-transform infrared (FTIR) vibrational spectrum of sesamin. The experimental data is compared with quantum chemical calculations of the most stable stereoisomers of sesamin to assign the experimental peaks following the methods used in other work on similarly-sized molecules [12, 13]. Computational spectra suggest that a combination of both hydrated and non-hydrated conformers populate the experimental spectrum.

## Materials and methods

### Experimental FTIR

Sesamin [[5,5′-[(1*S*,3a*R*,4*S*,6a*R*)-Tetrahydro-1*H*,3*H*-furo[3,4-*c*]furan-1,4-diyl]bis(2*H*-1,3 benzo-dioxole); 354.35 g/mol]] (Fig 1) powder was purchased from Sigma (SMB00705, 10mg, >98% purity HPLC) and used without further purification. Reported purity for this lot of sesamin was 98.9%. The sample was kept at -20˚C, then thawed at least three hours before any experimental manipulation. FTIR transmission measurements for sesamin were performed to identify vibrational modes in sesamin molecules at room temperature. A Nicolet IR100 FTIR spectrometer (Thermo-Fisher Scientific) was used with a deuterated triglycine sulfate (DTGS) detector. Sesamin (0.5 mg) was mounted onto a KBr substrate because KBr is transparent to the FTIR spectral range. Spectra were scanned 256 times and averaged at room temperature. The baseline was corrected in the absorbance spectrum after conversion from transmission raw spectrum using antisymmetric least squares smoothing method. Wavenumbers below 500 cm$^{-1}$ and above 3500 cm$^{-1}$ were not considered in the peak assignment since signal was low compared to noise. Normalization of the raw data was performed by dividing all intensity values by the maximum value in the dataset, and intensity values were plotted against frequencies.

### Computational generation of sesamin structures and populations and calculation of IR spectra

Stereoisomers of sesamin were built using the program MOE [14], and utilizing the carbon numbering system described by Baures, *et al.* [15], (Fig 1). The C4, C1', C1, and C4' carbon atoms are chiral, thus the number of stereoisomers of sesamin is $2^4$ = 16. However, because of sesamin's structural symmetry, six of these stereoisomers are identical, leaving ten distinct

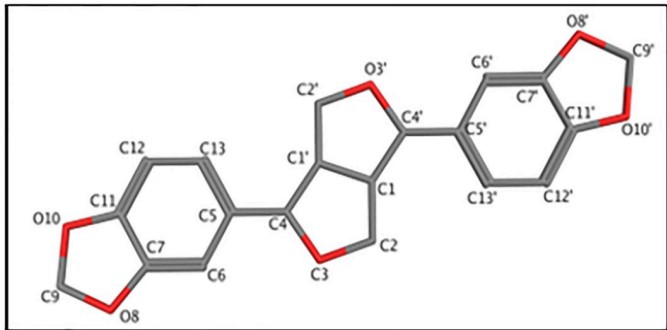

**Fig 1. Structure of sesamin.** Atom numbering is that used by Baures, *et al.*

isomers, as given in Fig 2. These molecules were energy minimized using the program Spartan'20 [16] at the EDF2/6-31G* level of theory, appropriate for IR spectra calculations [17]. The FTIR spectra were calculated for every sesamin species, listed in Fig 2 (including structurally redundant species), and for different levels of hydration (as described in the section below), using the FTIR calculation facility in Spartan.

The IR spectra from different species were combined by weight according to the species' relative populations, calculated from the species' relative energies using the Boltzmann population ratio equation:

$$Nb/Na = e(-\Delta E/RT) \tag{1}$$

where $Nb$ and $Na$ are the relative populations of two sesamin species, $\Delta E$ is the relative energy difference between these species, and RT = 0.6 kcal/mol at room temperature. Normalization of data sets was performed by dividing all intensity values by the maximum value in the dataset. As for the experimental spectrum, the combined normalized and weighted calculated spectra were plotted against the normalized experimental spectrum.

## Hydration of sesamin isomers

The experimental spectrum (see below) suggests the possibility of sesamin hydration in the "water band" in the ~1640 cm$^{-1}$ region [18]. Spartan'20 was therefore used to add one, two, and three water molecules to the minimized isomers of sesamin (Fig 3; doubly-hydrated example). Each water molecule was initially positioned within 2.5 Å of sesamin. Hydrated species were energy-minimized using the EDF2/6-31G* level of theory. Boltzmann population and IR spectra for these hydrated sesamin were calculated as described above.

## Results and discussion

### Physical FTIR measurements

FTIR measurements of sesamin powder were followed by conversion from the raw transmission spectrum to absorbance spectrum and baseline removal using the antisymmetric least squares smoothing method. The atmospheric $CO_2$ signal at 2350 cm$^{-1}$ (not included in the spectral range in this paper) was removed by taking the mean value from adjacent data points before baseline removal. The FTIR absorption spectrum was examined from the range of 580 cm$^{-1}$ to 2000 cm$^{-1}$. Fig 4 shows the IR spectrum of sesamin, with major peaks' wavenumbers labeled and numbered from one to 15 for comparison purposes. Shoulder-like features which can be found in peaks 12 and 13 are not assigned as separate peaks in this spectrum as peak deconvolutions are challenging and could provide inaccurate information [19].

### IR spectra assignments

The computed FTIR spectrum for sesamin was calculated as described in Methods. The energy minimization and Boltzmann-weighted population calculations of sesamin isomers suggest that more than one single species is present in the sample at room temperature (Table 1). The top five species listed in Table 1 account for more than 99.5% of the predicted sesamin population.

Each individual IR spectrum calculated from Spartan'20 was combined and weighted according to stereoisomers' populations given in Table 1, then normalized. The resulting spectrum is compared to the normalized experimental spectrum in Fig 5, with labels assigned as given in Table 2. Frequencies in the region of 580 to 2000 cm$^{-1}$ are analyzed as this frequency

| Isomer Number | Stereoisomer (without benzodioxol side chains) | | Isomer Number | Stereoisomer (without benzodioxol side chains) | |
| --- | --- | --- | --- | --- | --- |
| 1 |  4R, 1'R, 1S, 4'S |  4S, 1'S, 1R, 4'R | 6 |  4S, 1'S, 1S, 4'S | |
| 2 |  4R, 1'S, 1R, 4'R |  4R, 1'R, 1S, 4'R | 7 |  4R, 1'S, 1S, 4'S |  4S, 1'S, 1S, 4'R |
| 3 |  4S, 1'S, 1R, 4'S |  4S, 1'R, 1S, 4'S | 8 |  4R, 1'R, 1R, 4'S |  4S, 1'R, 1R, 4'R |
| 4 |  4R, 1'S, 1R, 4'S |  4S, 1'R, 1S, 4'R | 9 |  4R, 1'S, 1S, 4'R | |
| 5 |  4R, 1'R, 1R, 4'R | | 10 |  4S, 1'R, 1R, 4'S | |

**Fig 2. Stereoisomers of sesamin.** Benzodioxol side chains are omitted. Stereoisomers are numbered according to unique configurations. Table includes structurally redundant species.

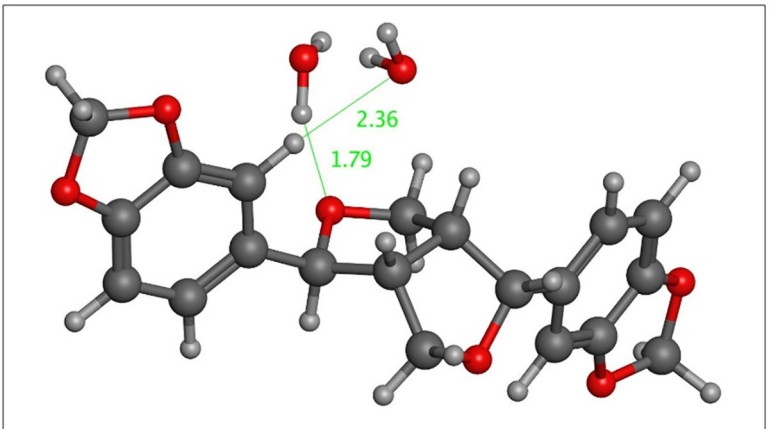

**Fig 3. Isomer 8 hydrated with two water molecules, each located within 2.5 Å of sesamin.** Bond distances (in Å) are indicated in green.

domain represents the most active spectral region for related lipid soluble lignans of sesame oil [20].

Individual isomers' calculated spectra are given in S1–S5 Figs. In the following discussion, the comparison between the experimental and calculated spectra was based on the combined non-hydrated spectra built as described above from the top five compounds given in Table 1 and shown in Fig 5.

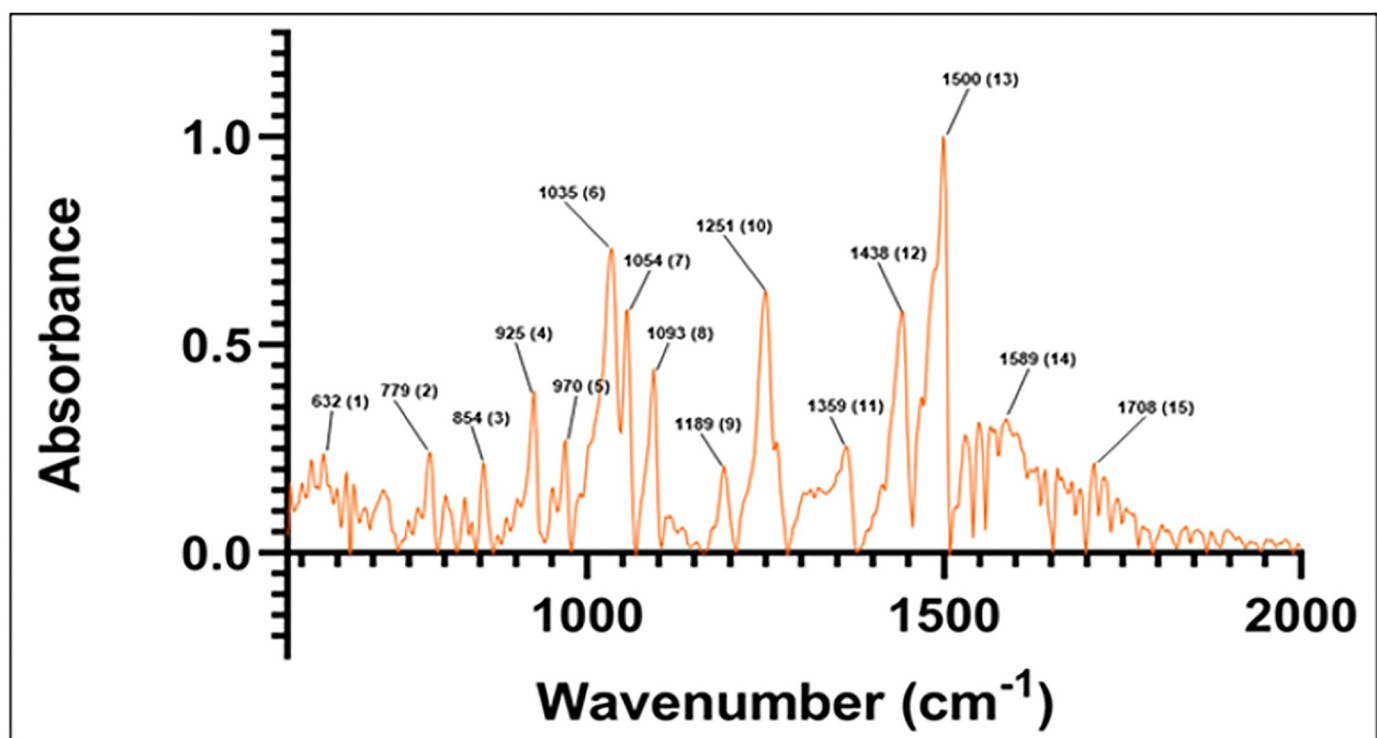

**Fig 4. Experimental FTIR spectrum of sesamin.** Numbered peak frequencies are indicated in Table 2 below.

**Table 1. Calculated energies and population ratios of stereoisomers of sesamin.**

| Energies and Relative Populations of Stereoisomers of Sesamin | | | |
|---|---|---|---|
| Isomer (#s from Fig 2) | Energy (Hartrees) | Population Ratio | Percent |
| 8 | -1223.43836 | 1.00 | 40.21% |
| 7 | -1223.43830 | 0.94 | 37.72% |
| 6 | -1223.43701 | 0.24 | 9.76% |
| 5 | -1223.43692 | 0.22 | 8.91% |
| 10 | -1223.43587 | 0.07 | 2.97% |
| 9 | -1223.43401 | 0.01 | 0.42% |
| 4 | -1223.41913 | 1.85E-09 | 7.45E-10% |
| 3 | -1223.41840 | 8.58E-10 | 3.45E-10% |
| 2 | -1223.41837 | 8.31E-10 | 3.34E-10% |
| 1 | -1223.41795 | 5.37E-10 | 2.16E-10% |

Stereoisomers were minimized using the EDF2 level of theory. Isomer numbers correspond to those listed in Fig 2.

Each of the species in the calculated spectrum individually demonstrates major peaks in the range of 1000–1500 cm$^{-1}$, with minor peaks between 1300 and 1450 cm$^{-1}$. A vibrational analysis of these individual major peaks reveals vibrations across all species which can be grouped according to similarities in peak frequency ranges, as indicated in Table 3.

The calculated major peak locations are used to assign the experimental major peaks (those within 6 cm$^{-1}$ of the computed peaks), as indicated in Table 2. In particular, peak 8 (which represents peaks in the range 1092–1100 cm$^{-1}$ from Table 2 and S1 Video), peak 10 (1318–1342

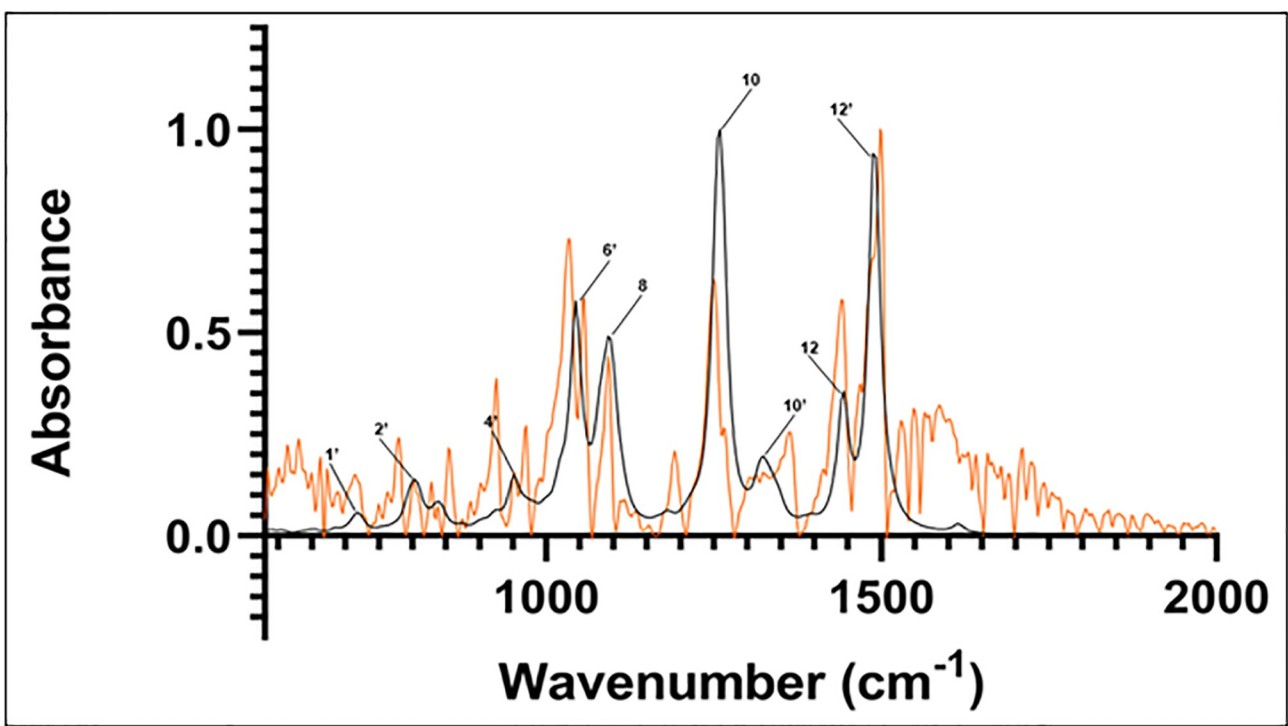

**Fig 5. Calculated spectrum of sesamin isomers (black).** Experimental spectrum (orange). Major peaks' frequencies are labeled according to numbering in Table 2.

**Table 2. Major peak frequencies of experimental and top sesamin species, *in vacuo* and hydrated.**

| Peak Number | Major Peak Frequencies (cm⁻¹) of Experimental Sesamin | Major Peak Frequencies (cm⁻¹) for Most-Populated Computed Species | | | | | | | | |
|---|---|---|---|---|---|---|---|---|---|---|
| | | *In vacuo*: | | | | | | Hydrated: | | |
| | | 8 | 7 | 6 | 5 | 10 | Combined | 9 +1 water: | 8 +2 waters: | 6 +3 waters: |
| 1 | 632 | | | | | | | | 691 | 665 |
| 1' | | 722 | 711 | | | | 724 | | | |
| 2 | 779 | | | | | 788 | | | | |
| 2' | | 803 | 803 | 803 | 802 | 806 | 808 | 807 | | |
| 3 | 854 | 844 | | | | | | | 851 | 898 |
| 4 | 925 | | | | | | | | | |
| 4' | | 953 | 950 | 951 | 952 | 954 | 952 | 955 | 953 | |
| 5 | 970 | | | | | | | | | |
| 6 | 1035 | | | | | | | | | 1025 |
| 6' | | 1044 | 1044 | 1043 | 1043 | 1043 | 1044 | 1041 | 1039 | 1043 |
| 7 | 1054 | | | | | | | | | |
| 7' | | 1080 | 1081 | | | 1083 | | 1066 | 1075 | 1080 |
| 8 | 1093 | 1092 | 1092 | 1100 | 1100 | | 1092 | | | |
| 9 | 1189 | | | | | | | | | |
| 10 | 1251 | 1259 | 1256 | 1254 | 1254 | 1263 | 1260 | 1256 | 1262 | 1267 |
| 10' | | 1322 | 1318 | 1327 | 1326 | 1336 | 1324 | 1321 | 1324 | |
| 11 | 1359 | | | | | | | | | |
| 12 | 1438 | 1443 | 1445 | 1441 | 1441 | 1444 | 1444 | 1447 | 1443 | 1445 |
| 12' | | 1491 | 1489 | 1489 | 1488 | 1488 | 1488 | 1488 | 1490 | 1493 |
| 13 | 1500 | | | | | | | | | |
| 14 | 1589 | | | | | | | | | |
| 15 | 1708 | | | | | | | | | |

Each major peak is assigned a number based on position relative to those in the experimental spectrum. Isomer numbers correspond to those listed in Fig 2.

cm⁻¹ from Table 2 and S2 Video), peak 12 (1440–1445 cm⁻¹ from Table 2 and S3 Video), and peak 13 (1485–1491 cm⁻¹ from Table 2 and S4 Video) exhibit intensity and frequency similarity. Peaks 8 and 13 share close locations with the assigned peaks in the experimental FTIR spectrum of sesamol, a related lignan component of sesame oil [20]. An additional peak in the sesamol spectrum aligns near to the peak at 1043–1046 cm⁻¹ in the computed sesamin spectrum. The computed sesamin peak exhibits vibrations like that assigned to sesamol (S5 Video).

**Table 3. Motions in major peaks of the most populous sesamin species, grouped by major frequency ranges.**

**Peak Assignments for Sesamin from Computed FTIR**

| Peak (cm-1) | Assignment(s) | Location(s) |
|---|---|---|
| 1043–1046 | antisymmetric stretching | in-plane of dioxolanes, benzenes |
| 1092–1100 | antisymmetric stretching, rocking | global distribution |
| 1253–1263 | rocking, antisymmetric stretching | within benzenes and their hydrogens; all other bonds (minor) |
| 1318–1342 | rocking | furans, benzenes (minor) |
| 1440–1445 | rocking, antisymmetric stretching | benzenes, oxolanes, C4 hydrogens |
| 1485–1491 | antisymmetric stretching, scissoring, rocking | within benzenes, oxolanes, C2 hydrogens |

**Fig 6. Vibrational mode of isomers *in vacuo* at 1044 cm$^{-1}$ (cyan arrows).**

Vibrations in this region associated with sesamol's phenolic hydroxyl group unsurprisingly do not appear in the sesamin spectrum.

The vibrations at these peaks' frequencies are generally similar to one another. This includes C-H rocking in benzenes, C-O antisymmetric stretching and C-H wagging in the dioxolanes, and antisymmetric stretching and rocking within the furans. Symmetric stretching and other bending motions such as twisting and scissoring are largely not observed. In addition, owing to sesamin's symmetry, vibrations that correspond to flanking regions of a peak (within a few cm$^{-1}$) are generally a mirror image of one another. These vibrations also straddle the furanofuran line of symmetry within the molecule. For example, calculated vibrations at 1044 cm$^{-1}$ consist of C-O antisymmetric stretch of one oxolane and C-H rocking on the C6 and C12 atoms of its neighboring benzene (Fig 6). We see the same vibration but on the opposite side of the molecule at neighboring frequency 1046 cm$^{-1}$, which, along with the 1044 cm$^{-1}$ resonance, comprises major peak 6' (Table 2).

Other calculated peaks do not exhibit such clear close frequencies to that of the experimental sesamin peaks. For example, calculated peak number 6' is located between experimental peaks number 6 and 7; experimental and calculated peaks 11 and 10', respectively, have no equivalent to each other; and experimental peak number 13 at 1500 cm$^{-1}$, the most intense peak in the experimental spectrum, appears to contain two other minor peak shoulders at 1469 cm$^{-1}$ and 1488 cm$^{-1}$, while calculated peak number 12', at 1488 cm$^{-1}$, is located in between these experimental peaks (Fig 5; Table 2).

## Hydration

As the experimental FTIR data collection was performed in open air at room temperature, and because of its IR signal in the "water region" of ~1640 cm$^{-1}$, there may exist a partial hydration of the sample. This would suggest a different calculated spectrum than the one presented in Fig 5, based only on *in vacuo* sesamin isomers. As described in Methods, sesamin species from the *in vacuo* study were each hydrated with one, two, or three water molecules. The population analysis of these hydrated sesamin systems is given in Table 4.

In all cases, the most populous sesamin species in the presence of water was found to represent more than 95% of the population. Hence, only these species' calculated spectra were used to compare to the experimental spectrum. The spectra from these calculations can be seen in Figs 6–8 (isomer 9 + 1 water, isomer 8 + 2 waters, and isomer 6 + 3 waters, respectively) superimposed onto the experimental spectrum. Notably, isomer 9 in this group exhibits possible stabilization due to the presence of water; it was not present in the top five *in vacuo* calculations. In all plots, a minor peak at ~1640 cm$^{-1}$ represents resonance of pure water [21].

**Table 4. Population proportions of hydrated sesamin species.**

| Population Proportions of Hydrated Sesamin Species | | | | | |
|---|---|---|---|---|---|
| Isomer # | 1 water | Isomer # | 2 waters | Isomer # | 3 waters |
| 9 | 95.74% | 8 | 99.91% | 6 | 99.998% |
| 8 | 1.17% | 6 | 0.04% | 10 | 8.83E-06% |
| 7 | 0.01% | 7 | 1.95E-06% | 8 | 5.03E-07% |
| 6 | 2.05E-05% | 5 | 1.07E-06% | 2 | 6.10E-08% |
| 10 | 1.10E-04% | 9 | 5.94E-07% | 7 | 9.14E-11% |
| 5 | 6.21E-05% | 10 | 8.52E-10% | 5 | 8.28E-11% |
| 3 | 2.90E-12% | 2 | 4.25E-14% | 1 | 7.18E-15% |
| 4 | 4.57E-13% | 1 | 1.62E-17% | 9 | 9.08E-16% |
| 2 | 1.38E-13% | 3 | 5.72E-18% | 4 | 6.29E-23% |
| 1 | 1.35E-13% | 4 | 2.13E-18% | 3 | 1.44E-23% |

Isomer numbers correspond to those listed in Fig 2.

The correspondence between experimental and *in vacuo* calculated peaks 10 and 12 is preserved in the three calculated hydrated spectra. The location of calculated peak 12' and its close vicinity to experimental peak number 13 is also observed in the calculated hydrated spectra. That calculated peak is still a single peak; adding hydration to the structure and calculations does not lead to any deconvolution into several close peaks that could be used to assign the satellite vibrations close to experimental peak 13.

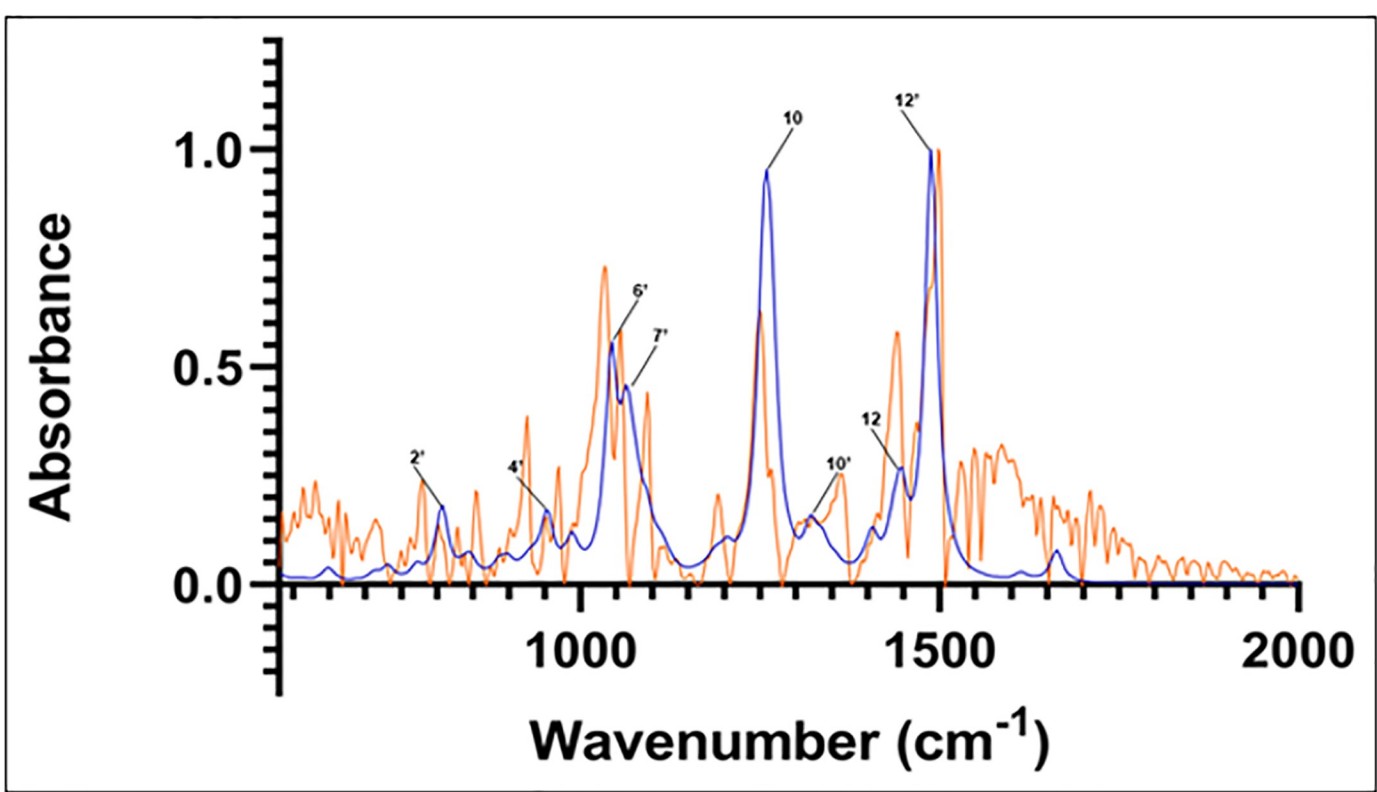

**Fig 7. Computed spectrum of isomer 9 +1 water (blue) and experimental IR spectrum (orange).** Refer to Table 2 for peak frequency values.

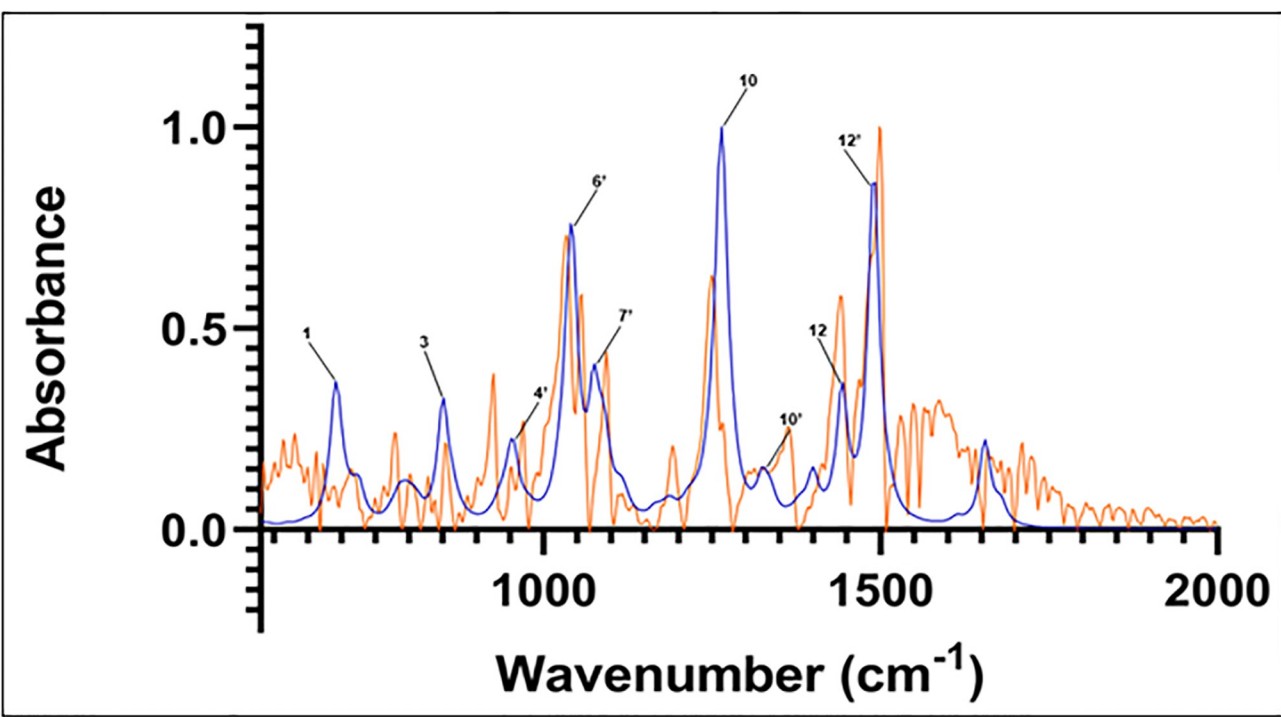

**Fig 8. Computed spectrum of isomer 8 +2 waters (blue) and experimental IR spectrum (orange).** Refer to Table 2 for peak frequency values.

The main effects of sesamin hydration on the IR calculated spectra are seen in the 1000 cm$^{-1}$–1100 cm$^{-1}$ region, corresponding to experimental peaks 6, 7, and 8. In that region, the effect of hydration appears to be altering the *in vacuo* calculated spectrum in a significant way, but differently depending on the number of water molecules interacting with sesamin. In the case of hydration with one water molecule (Fig 7), calculated peak 8 is reduced to a shoulder and peak 7' appears. That could correspond to the deconvolution of calculated peak 6' into two peaks that are equivalent to experimental peaks 6 and 7. In general, the vibrations we see associated with peaks 6' and 7' in this calculated hydrated spectrum match those at corresponding frequencies of the *in vacuo* spectrum. This applies as well for vibrations associated with peak 8 *in vacuo* and the same frequency in the hydrated species, where no obvious peak is observed.

Within that same region, in the case of doubly-hydrated sesamin (Fig 8), calculated peak 6' at 1044 cm$^{-1}$ has the same vibrations as 1044 cm$^{-1}$ *in vacuo*, however, these are different from vibrations at 1039 cm$^{-1}$ in the hydrated species. Peak 7', located between experimental peaks 7 and 8, exhibits the same vibration as 1075 cm$^{-1}$ *in vacuo*, which is between peaks. Likewise, vibrations at 1089 and 1093 cm$^{-1}$ in the hydrated spectrum have the same motions as the peak 8 (which consists of frequencies 1092 cm$^{-1}$ and 1094 cm$^{-1}$) *in vacuo* vibrations.

For these hydrated species, the vibrations appear to be identical or equivalent to those observed in the *in vacuo* calculated spectrum, but with different intensities. This suggests the possibility of water influencing sesamin's vibration by modulating the dipole moments' variation rather than the nature of the vibrations themselves.

The only noteworthy exception to this is in the isomer minimized with three water molecules (Fig 9). Peak 6 vibrations (see S6 Video) in this spectrum do not mimic those at the corresponding registered frequency of 1021 cm$^{-1}$ of the *in vacuo* spectrum (S7 Video). This is a new peak (Fig 10). In addition, the vibrations corresponding to the frequencies of peak 6' (in

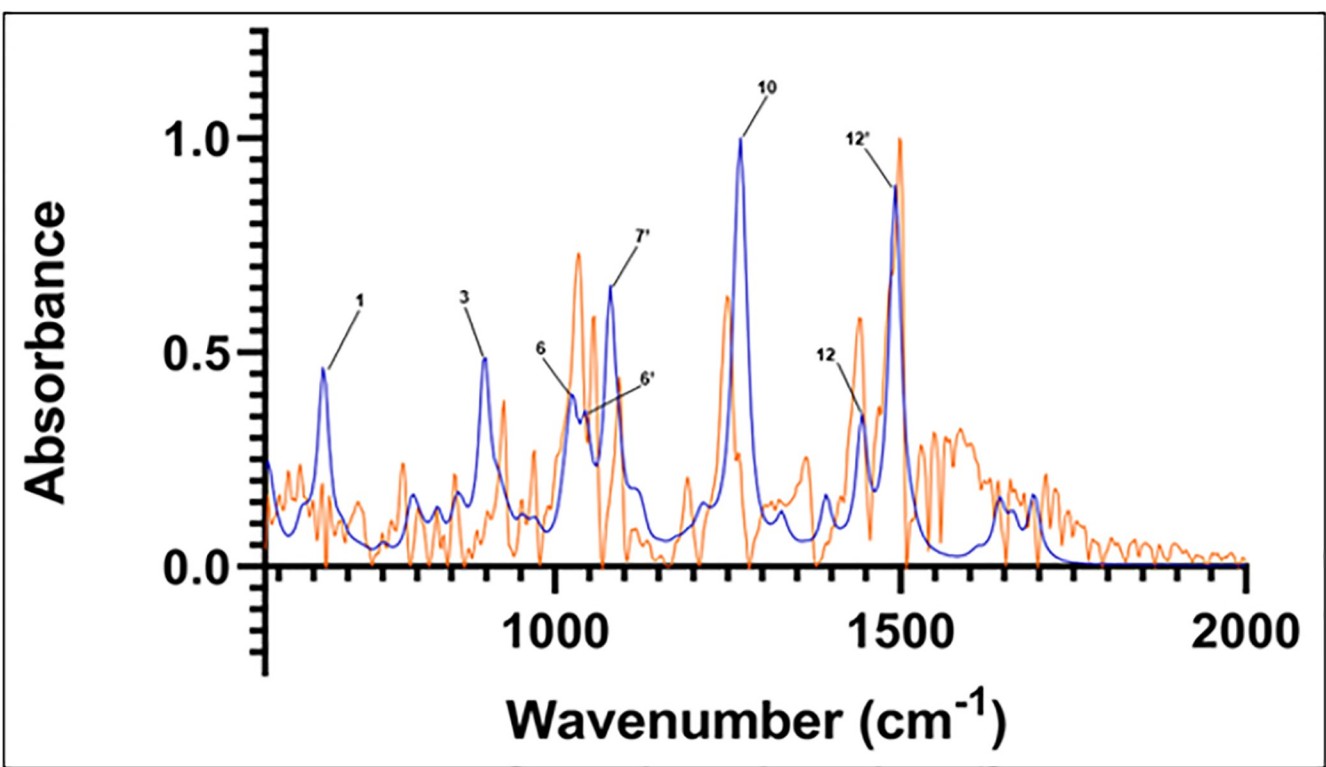

**Fig 9. Computed spectrum of isomer 6 +3 waters (blue) and experimental IR spectrum (orange).** Refer to Table 2 for peak frequency values.

the range of 1043–1046 cm$^{-1}$; S8 and S9 Videos) are similar, but with additional vibrations occurring in the furanofuran moiety of the hydrated species which the *in vacuo* species doesn't exhibit (Fig 10; S5 Video). Peaks 7' and 8 of this spectrum do, however, exhibit vibrations that are the same as those at the same frequencies *in vacuo*. In this case, although no vibrations

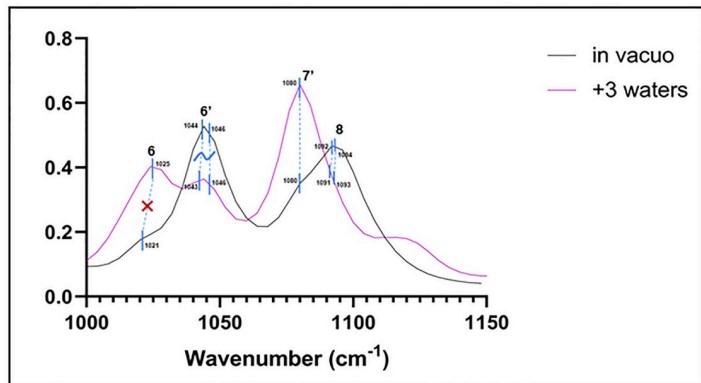

**Fig 10. Peaks of *in vacuo* (black) and triple-hydrated (pink) species of sesamin in the 1000–1150 cm$^{-1}$ frequency range.** Blue dotted lines connect frequencies across the two spectra which correspond to the same vibrations. The dotted line at peak 6 is marked with a red 'X' because vibrations between the two are not the same. The dotted lines at peak 6' are marked with a tilde to represent a combination of shared and different vibrations for these isomers. The Y axis represents absorbance.

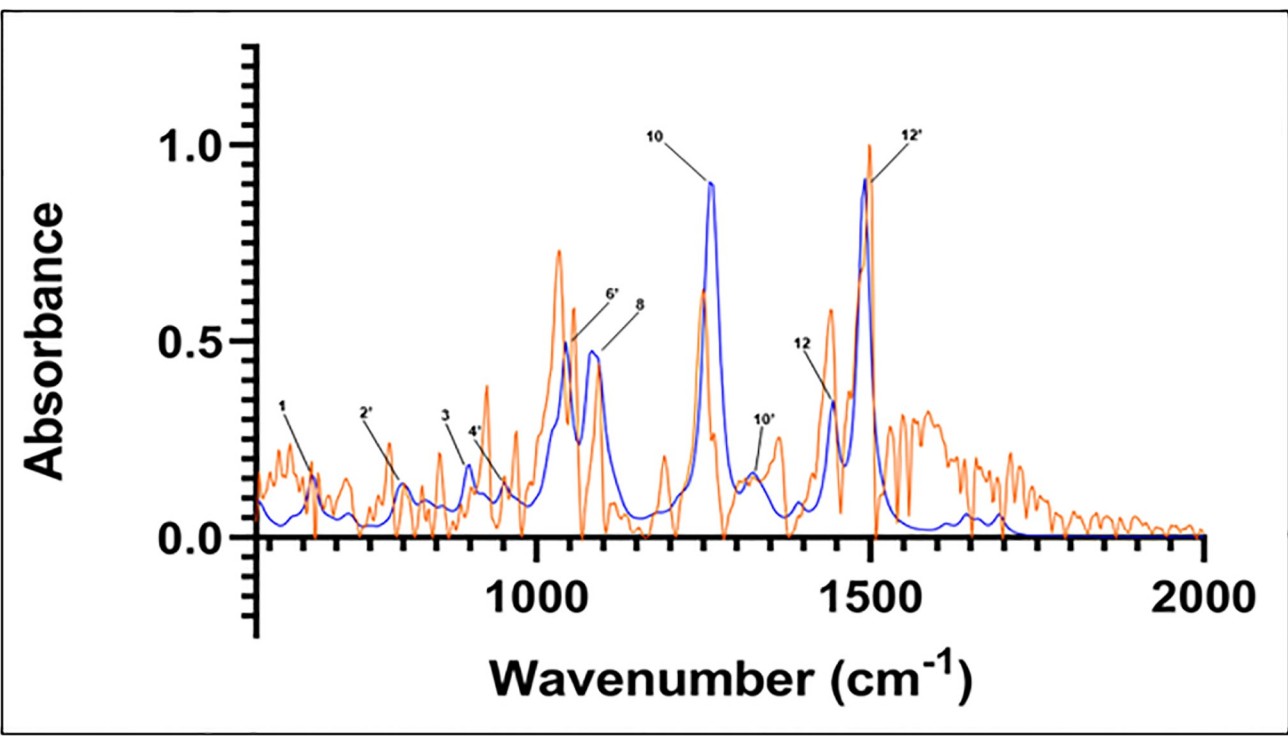

**Fig 11. Combination of IR spectra from isomer 8-*in vacuo*, isomer 7-*in vacuo*, and isomer 6 +3 water molecules (blue).** Experimental spectrum is in orange. Refer to Table 2 for peak frequency values.

within water molecules were observed at these frequencies, water molecules influence the frequencies of sesamin vibrations.

## Combination spectrum

An IR spectrum was calculated by including the top two most populated species from *in vacuo* calculations (isomers 8 and 7) along with the most populated triple-hydrated species (isomer 6). Unlike in calculations for species above, weights were not assigned to these combined species. This spectrum is compared to that of the experimental in Fig 11. Here we observe similar correspondence of major peaks between 1200 and 1500 cm$^{-1}$ as in the previous comparisons. As with the *in vacuo* calculated spectrum, we observe a peak between those of 1039 and 1054 cm$^{-1}$ (corresponding to peak 6', Table 2) in the experimental spectrum. However, the calculated peak at ~1093 cm$^{-1}$ in this combination spectrum is redshifted and exhibits a wider absorbance band compared to both experimental and *in vacuo* peak 8.

## Conclusions

In this work, we have collected an experimental Fourier-transform infrared spectrum of sesamin. This was followed by a comparison of this spectrum with a computationally determined FTIR spectrum of the top stereoisomers of sesamin *in vacuo*. The computational calculations were used to assign previously undefined vibrational modes to many of sesamin's major peaks. Discrepancies between experimental and computed spectra were subsequently investigated in the context of sesamin hydration. Though sesamin is not water soluble, the nature of experimental sample collection lends itself to some level of ambient hydration which may account

for differences in peak frequencies. FTIR calculations of sesamin species with one, two, and three molecules of water were consecutively computed. In previously published spectroscopic characterizations of aliphatic ethers and other small molecules, also using Boltzmann weighting of isomers as performed here, a difference between the experimental and theoretical vibrational peaks with margins similar to those recorded in our study is observed [12, 13]. Differences of up to 132 cm$^{-1}$ were reported by Bec, *et al*, in ethanol between experimental and calculated bands [22]. In the present work, the largest difference between Boltzmann-weighted combination of individual spectra and experimental spectra is 92 cm-1 (Table 2 and S1 Table), and the average difference is around 25 cm-1 (S1 Table).

In general, motions at major peaks include antisymmetric stretching and rocking in the benzene and dioxolane groups. These vibrations, however, become more global across all sesamin groups (including in the furan rings) following hydration, and some peaks of the spectrum demonstrate notable shift. For each of the calculations computed, no spectrum peaks were observed, which is in perfect agreement with the experimental spectrum. The triply-hydrated spectrum is qualitatively more like the experimental spectrum due to the split in the 6' peak at ~1020–1050 cm$^{-1}$ and indicates that residual hydration of the sesamin powder exists. Further calculations of hydrated sesamin may determine more precisely the level of ambient hydration to match the peaks of the experimental spectrum more precisely. Albeit higher-resolution spectra and higher-level calculations will be interesting to obtain and perform in the future, the current findings contribute to the characterization of sesamin beyond the existing basic descriptions and solidify quantum minimization methods as useful spectral interpretations of small molecules.

## Supporting information

**S1 Fig. Experimental and computed isomer 8 sesamin spectra.** Experimental spectrum (orange); *in vacuo* isomer 8 (blue).
(TIF)

**S2 Fig. Experimental and computed isomer 7 sesamin spectra.** Experimental spectrum (orange); *in vacuo* isomer 7 (blue).
(TIF)

**S3 Fig. Experimental and computed isomer 6 sesamin spectra.** Experimental spectrum (orange); *in vacuo* isomer 6 (blue).
(TIF)

**S4 Fig. Experimental and computed isomer 5 sesamin spectra.** Experimental spectrum (orange); *in vacuo* isomer 5 (blue).
(TIF)

**S5 Fig. Experimental and computed isomer 10 sesamin spectra.** Experimental spectrum (orange); *in vacuo* isomer 10 (blue).
(TIF)

**S1 Video. Resonance of *in vacuo* isomer 8 at frequency 1092 cm$^{-1}$.**
(MOV)

**S2 Video. Resonance of *in vacuo* isomer 8 at frequency 1259 cm$^{-1}$.**
(MOV)

**S3 Video. Resonance of *in vacuo* isomer 8 at frequency 1443 cm$^{-1}$.**
(MOV)

**S4 Video. Resonance of *in vacuo* isomer 8 at frequency 1491 cm$^{-1}$.**
(MOV)

**S5 Video. Resonance of *in vacuo* isomer 8 at frequency 1044 cm$^{-1}$.**
(MOV)

**S6 Video. Resonance of triply-hydrated isomer 6 at frequency 1025 cm$^{-1}$.**
(MOV)

**S7 Video. Resonance of in vacuo isomer 8 at frequency 1021 cm$^{-1}$.**
(MOV)

**S8 Video. Resonance of triply-hydrated isomer 6 at frequency 1043 cm$^{-1}$.**
(MOV)

**S9 Video. Resonance of triply-hydrated isomer 6 at frequency 1046 cm$^{-1}$.**
(MOV)

**S1 Table. Differences between selected experimental and combined in vacuo calculated major peak frequencies.**
(TIF)

## Author Contributions

**Conceptualization:** Moon-Hyung Jang, C. Ryan Yates, Joseph Ng, Jerome Baudry.

**Data curation:** Sara W. Jackson, Eliza Asani, Joseph Ng.

**Formal analysis:** Sara W. Jackson, Moon-Hyung Jang, Eliza Asani, Joseph Ng, Jerome Baudry.

**Investigation:** Sara W. Jackson, Moon-Hyung Jang, Joseph Ng.

**Methodology:** Sara W. Jackson, Moon-Hyung Jang, Joseph Ng, Jerome Baudry.

**Project administration:** Jerome Baudry.

**Software:** Jerome Baudry.

**Supervision:** Joseph Ng, Jerome Baudry.

**Validation:** Sara W. Jackson, Moon-Hyung Jang, Eliza Asani, C. Ryan Yates, Joseph Ng, Jerome Baudry.

**Visualization:** Sara W. Jackson, Eliza Asani, Joseph Ng.

**Writing – original draft:** Sara W. Jackson, Moon-Hyung Jang, C. Ryan Yates, Joseph Ng, Jerome Baudry.

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
