## [Decision Letter · Decision Letter 0]

8 Jan 2024

PONE-D-23-41995Near infrared spectroscopic characterization of sesamin, a dietary lignan natural productPLOS ONE

Dear Dr. Baudry,

Thank you for submitting your manuscript to PLOS ONE. After careful consideration, we feel that it has merit but does not fully meet PLOS ONE’s publication criteria as it currently stands. Therefore, we invite you to submit a revised version of the manuscript that addresses the points raised during the review process.

 The referees have made some constructive comments (which may be found below) that would need to be addressed in order for this manuscript to be considered suitable for publication. One of the referees has indicated that the selected methods might not necessarily be entirely appropriate in the context of the present study, and has indicated that alternative calculations might be more appropriate in order to provide a more thorough analysis. The other referee has indicated that further analysis would be required, and that consideration of solvation effects etc... should also be taken into consideration.

We look forward to receiving your revised manuscript.

Kind regards,

Robert John O'Reilly, Ph.D.

Academic Editor

PLOS ONE

Journal Requirements:

3. We note that your Data Availability Statement is currently as follows: "All relevant data are within the manuscript and its Supporting Information files."

Reviewers' comments:

Reviewer's Responses to Questions

**Comments to the Author**

1. Is the manuscript technically sound, and do the data support the conclusions?

Reviewer #1: Partly

Reviewer #2: Yes

2. Has the statistical analysis been performed appropriately and rigorously? 

Reviewer #1: Yes

Reviewer #2: Yes

3. Have the authors made all data underlying the findings in their manuscript fully available?

Reviewer #1: Yes

Reviewer #2: Yes

4. Is the manuscript presented in an intelligible fashion and written in standard English?

Reviewer #1: Yes

Reviewer #2: Yes

5. Review Comments to the Author

Reviewer #1: The authors have attempted to use IR to elucidate the different stereoisomers of sesamin. It should be noted that IR alone may not be sufficient to definitively identify stereoisomers. While IR spectroscopy provides valuable information about functional groups and molecular vibrations, it might not distinguish between stereoisomers that have the same atoms and functional groups but differ in their three-dimensional arrangement. NMR spectroscopy, specifically using chiral solvents or chiral shift reagents, can also provide information about the configuration of a chiral center by observing the splitting patterns or chemical shifts of surrounding atoms. Chromatographic techniques with chiral stationary phases and polarimetry are also used for chiral analysis, although they might not directly determine the configuration of a chiral carbon but can help identify enantiomers. Thus, it is better to do some NMR calculation in these molecules instead. The basis set also that they used is too small to distinguish such changes. As such, this paper is not suitable for publication in its present state.

There are some references that are excluded such as

https://doi.org/10.1080/10408398.2021.1881438

https://doi.org/10.1016/j.bpc.2004.10.001

Reviewer #2: This manuscript investigates sesamin, a lignan in sesame seed oil, by Fourier-transform infrared spectroscopy. The study compares the experimental spectrum with quantum chemical calculations of sesamin's stereoisomers, revealing potential hydrated and non-hydrated conformers. The findings pave the way for a comprehensive understanding of its structure and potential applications in pharmacology.

The work general employs correct methods and follows the right principles.

That being said, the employed methods are largely straightforward and part of the discussion presented in the manuscript should be approached with reserve. While credit may be given to the effort, the apparent in-depth discussion of the variations in the theoretical spectra of complexes with different number of water molecules are largely speculative (or, "academic"). In vacuo calculations for such systems are still only roughly representative to a solid/hydrated sample. This should be better reflected in the manuscript to prevent misleading readers not being experts in this area.

Nonetheless, the collected results and interpretations provide useful contribution to the field and I find this work publishable, provided a minor revision is considered by the authors.

1. The title and the abstract ("This work provides the first experimental near infrared spectrum of sesamin") both specifically mentions NIR region, yet NIR spectrum is not presented in this work at all. Instead, the reader is informed that NIR region is not interesting, i.e., "Wavenumbers below 500 cm-1 (and above 3500 cm-1) were not considered in the peak assignment since signal was low compared to noise.

2. B3LYP/6-31G* by contemporary standards should not be considered "appropriate for IR spectra calculations". But, judging from the figures shown in this work, it’s sufficient in the presented case.

3. What are the symmetries of each of the considered structures? If they differed by the number of symmetry elements, then the Eq. 1 should be refined with the use of symmetry cofactors (see, e.g. DOI: 10.3390/molecules24112189).

4. Molecular vibrations are 'antisymmetric' not 'asymmetric'.

5. Experimental FTIR spectra are of poor quality (noisy, distorted baseline with negative peaks, deformed bands), which is difficult to understand, as the measurements should be fairly straightforward.

6. Previous works modelling IR spectra on the basis of Boltzmann-weighted spectra of co-existing forms, done using anharmonic calculations and yielding highly accurate theoretical lineshape should be cited (e.g.:, 10.1021/acs.jpca.6b11734, 10.1038/s41598-019-53827-6, or 10.1016/j.molliq.2014.02.028).

6. PLOS authors have the option to publish the peer review history of their article (what does this mean?). If published, this will include your full peer review and any attached files.

Reviewer #1: No

Reviewer #2: No

---

## [Decision Letter · Decision Letter 1]

8 May 2024

PONE-D-23-41995R1Infrared spectroscopic characterization of sesamin, a dietary lignan natural productPLOS ONE

Dear Dr. Baudry,

Thank you for submitting your manuscript to PLOS ONE. After careful consideration, we feel that it has merit but does not fully meet PLOS ONE’s publication criteria as it currently stands. Therefore, we invite you to submit a revised version of the manuscript that addresses the points raised during the review process.

We look forward to receiving your revised manuscript.

Kind regards,

Robert John O'Reilly, Ph.D.

Academic Editor

PLOS ONE

Journal Requirements:

**Additional Editor Comments:**

The modifications made to the manuscript have been accepted by the majority of the referees. There is one comment, which was provided by an additional referee who reviewed the revised version of the manuscript, that should be addressed (refer to the comment by Referee 4). Once this has been sufficiently addressed, the manuscript will be ready for acceptance.

Reviewers' comments:

Reviewer's Responses to Questions

**Comments to the Author**

1. If the authors have adequately addressed your comments raised in a previous round of review and you feel that this manuscript is now acceptable for publication, you may indicate that here to bypass the “Comments to the Author” section, enter your conflict of interest statement in the “Confidential to Editor” section, and submit your "Accept" recommendation.

Reviewer #1: (No Response)

Reviewer #2: All comments have been addressed

Reviewer #3: (No Response)

Reviewer #4: (No Response)

2. Is the manuscript technically sound, and do the data support the conclusions?

Reviewer #1: Partly

Reviewer #2: Yes

Reviewer #3: Yes

Reviewer #4: Yes

3. Has the statistical analysis been performed appropriately and rigorously? 

Reviewer #1: No

Reviewer #2: Yes

Reviewer #3: N/A

Reviewer #4: Yes

4. Have the authors made all data underlying the findings in their manuscript fully available?

Reviewer #1: No

Reviewer #2: Yes

Reviewer #3: Yes

Reviewer #4: Yes

5. Is the manuscript presented in an intelligible fashion and written in standard English?

Reviewer #1: Yes

Reviewer #2: Yes

Reviewer #3: Yes

Reviewer #4: Yes

6. Review Comments to the Author

Reviewer #1: The authors have given their comments to the reviewers, unfortunately, I believe the topic that they present that using IR to determine the isomers is not enough. In addition, the authors have utilized sub-par calculation methods with the size of the molecules and the existing capacity now of the workstations they need to increase their theoretical methodology. Specifically,

1) The basis sets used is very small and they do not implement any scaling factor. In addition, anharmonic frequencies need to be included to get the overtones and combination bands that could arise from different isomeric forms.

2) How did they choose the location of the water molecules?

3) Never present energies in Hartrees, need to convert it into either kJ or kcal.

4) Based on the answer to comments

- "The 2021 paper by Majdalawieh, et al, is already cited by us (Ref #4) in the second sentence of our manuscript, where we introduce sesamin and the (limited) vibrational work available". This is a review paper and there is no mention about IR. Indeed, you can't hardly find papers regarding isomeric forms just merely using IR.

- Their goal is to "give a global “IR fingerprint” of sesamin, in several possible states and structures" again this fingerprint region is not conclusive given the theoretical methodology they utilized.

Reviewer #2: The authors' revisions were carried out exhaustively and addressed all points raised in the previous round of peer-review. The manuscript is recommended to be accepted in its present form.

Reviewer #3: I have reviewed Revision 1 of this manuscript.

The main claims of this study were that previously unassigned peaks from experimental FTIR of sesamin were able to be assigned with the support of EDF2/6-31G* computed FTIR spectra of sesamin's most prevalent stereoisomers. Further, to account for ambient hydration of sesamin samples, various states of hydration were modelled computationally. General agreement between the computational and experimental spectra were reported, within the bounds of previously reported error for organic molecules. The methodolgy used was appropriate and the authors have provided a clear presentation and discussion of results, and reached a measured conclusion that does not overstate the outcome.

The introduction of the manuscript oriented this study in the existing literature, highlighting its pharmocolgical significance.

The supporting information is thorough and complete and the methodolgy provides sufficient detail for the reproduction of the experiments. The manuscript is generally well written in easy to parse English, with minor typographical errors.

As acknowledged in the conclusion, updated FTIR characterisation of sesamin using an expanded basis set for this computational method (as mentioned in Ref [16]) and higher resolution experimental FTIR spectra would be an important next step towards a more complete characteristion of sesamin.

The author has addressed the concerns of Reviewer 2 well, particularly improving the clarity of the manuscript in the aim and in setting expectations of the audience, significantly curtailing this review.

The current manuscript is satisfactory in meeting the scientific, technical and ethical standards required for publication in this journal.

Reviewer #4:  (ii) In the abstract the author claims that this study provides 'the first experimental infrared spectrum of sesamin.' Although the work is comprehensive in structural assignments of previously unassigned frequencies, I take exception that it represents the "first" spectroscopic assignment of sesamin. I refer you to https://doi.org/10.1016/j.saa.2019.117777, published in 2019 in Spectrochimica Acta Part A: Molecular and Biomolecular Spectroscopy where near-IR coupled with chemometric method was used in selective vibrational frequencies assignments of sesamin and sesamolin. I think the claim should be reevaluated but, regardless, it does contribute a wealth of data to important class of biological compounds.

7. PLOS authors have the option to publish the peer review history of their article (what does this mean?). If published, this will include your full peer review and any attached files.

Reviewer #1: No

Reviewer #2: No

Reviewer #3: No

Reviewer #4: No

---

## [Author Response · Author response to Decision Letter 1]

17 Jul 2024

Please see Cover Letter and revised manuscript for details on how we addressed Reviewer 4's comments, as requested by Dr. O'Reilly (editor)

---

## [Editor Report · Decision Letter 2]

18 Jul 2024

Infrared spectroscopic characterization of sesamin, a dietary lignan natural product

PONE-D-23-41995R2

Dear Dr. Baudry,

We’re pleased to inform you that your manuscript has been judged scientifically suitable for publication and will be formally accepted for publication once it meets all outstanding technical requirements.

Kind regards,

Robert John O'Reilly, M.D., Ph.D.

Academic Editor

PLOS ONE
---

## [Editor Report · Acceptance letter]

13 Aug 2024

PONE-D-23-41995R2 

PLOS ONE

Dear Dr. Baudry, 

I'm pleased to inform you that your manuscript has been deemed suitable for publication in PLOS ONE. Congratulations! Your manuscript is now being handed over to our production team.

Kind regards, 

on behalf of

Dr. Robert John O'Reilly 

Academic Editor

PLOS ONE